# Using Multivariate Outliers from Smartphone Sensor Data to Detect Physical Barriers While Walking in Urban Areas

**Ramona Ruiz Blázquez and Mario Muñoz-Organero *** 

Telematics Engineering Department, Carlos III University of Madrid, 28911 Madrid, Spain; raruizb@it.uc3m.es
* Correspondence: munozm@it.uc3m.es

**Abstract:** Nowadays, our mobile devices have become smart computing platforms, incorporating a wide number of embedded sensors such as accelerometers, gyroscopes, barometers, GPS receivers, and magnetometers. Smartphones are valuable devices for gathering user-related data and transforming it into value-added information for the user. In this study, a novel mechanism to process sensor data from mobile devices in order to detect the type of area the user is crossing while walking in an urban setting is presented. The method is based on combining outlier data analysis and classification techniques from data collected by several pedestrians while traversing an urban environment. A theoretical framework, composed of methods for detecting multivariate outliers combined with supervised classification techniques, has been proposed in order to identify different situations and physical barriers while walking. Each type of element to be detected is characterized by using a feature vector computed based on the outliers detected. Finally, a radial SVM is used for the classification task. The classifier is trained in a supervised way with data from 20 different segments containing several physical barriers and used later to assign a class to new un-labelled data. The results obtained with this approach are very promising with an average accuracy around 95% when detecting different types of physical barriers.

**Keywords:** multivariate outliers; machine learning; SVM; mobile sensor data

---

## 1. Introduction

An outlier is an observation that deviates so much from other observations as to arouse suspicions that it was generated by a different mechanism [1]. In some cases, they are the result of poorly calibrated data gathering sensors, incorrect data entry, or processing or coding errors. In other cases, outliers capture some novelty features and patterns in the data and represent valuable information that would be unnoticed if training a machine learning algorithm based on the entire dataset. Once detected, outliers could be eliminated from the dataset in order not to affect the data analysis or extracted from it in order to learn particular characteristics hidden inside special parts of the dataset. Outliers can be found in independent individual variables or grouped in clusters of related variables. Considering the number of variables, outliers are classified in univariate, bivariate or multivariate outliers, depending on the number of different components that make up the datasets under research. A multivariate outlier can be an outlier value due to a sudden error in one of its components, or by small systematic errors in several of them. A multivariate outlier can also capture some novelty features in the data and therefore be able to isolate elements of particular interest inside the data.

It can be said that the problem of outlier detection combines some aspects from the classification and clustering problems. In multivariate datasets, the task of detecting outliers represents one of the most important tasks when performing any data analysis, regardless of the domain or area of study,

since multivariate outliers can also appear as an extraordinary situation that allows discovering useful and valuable information that usually is hidden by the high dimensionality of the data.

Different techniques have been studied and presented for multivariate outlier detection in [2–7], such as statistical methods, principal component analysis, methods for searching projections, or methods based on data mining, including those based on distance and those based on local density, and clustering techniques.

The main purpose of this work is to be able to demonstrate that outliers, far from being a set of data to be thrown away, hide enormous potential that can be exploited with surprising results. Although a significant number of studies have been performed to improve outlier detection in order to identify anomalous data, for cleaning the dataset and improve the accuracy of machine learning models, still, little research has been conducted instead for studying multivariate outliers as a source of useful information.

In a previous work [8], we have already proposed a method to use multivariate outliers to identify traffic congestion situations, with a summary of the most common multivariate outlier detection methods. The idea was, instead of removing the outlier data from the dataset, to use it in order to characterize anomalous regions while driving. Multivariate outliers were detected with different techniques, including statistically robust methods as MCD (Minimum Covariance Determinant), density-based outlier detection methods as LOF (Local Outlier Factor) or OCSVM (One-Class Support Vector Machine).

Also, in [9], a novel algorithm based on DRNN (Deep Recurrent Neural Networks) for outlier detection in HAR (Human Activity Recognition) is introduced, proving the utility of working with outliers in the data.

Although there is a lot of research and publications where the different techniques for detecting multivariate outliers are studied and compared, there is little research, at least as far as we know, that deals with the use and application of these outliers as a main source of information, instead of the total data set, as is the main objective of this work.

In this paper, we propose a novel application for outliers from mobile sensor data in order to identify different situations while walking. In this case we have used SVM just for classification tasking instead of using it as a multivariate outlier detection technique as we introduced in [8].

The scientific novelty of this research is to demonstrate that multivariate outliers, hidden in the multidimensional data, far to be considered as sample errors, can be extraordinarily useful. For example, they can explain new relationships between variables not found before or even to make a scientific discovery, since these data can reach a 15% or 20% of the dataset and they are usually ignored for the majority of researchers. We can work with outliers as a source of secret knowledge waiting to be discovered.

The remainder of this paper is organized as follows. A review of related previous research is captured in Section 2. Section 3 presents the conducted procedures and methods describing the scenario used in order to gather the dataset. Section 4 describes the experimental results achieved and Section 5 shows the conclusions of this research.

## 2. Related Work

Data generated by wearable sensors are prone to different types of anomalies. Price sensitive, low cost and battery-powered wearable devices provide in many cases data streams in which sensed data is mixed with outliers of various kinds. In some cases, outlier detection is needed in order to remove random errors introduced in the data and therefore purify the data quality in order to better learn the intricate patterns in the data [10]. In other cases, outlier detection will be a valuable source of information in order to detect changes in behavior or in the underlying characteristics of the data [10].

In the line of removing outliers from wearable sensors, the authors in [10] provided a review of different methods that have been used for outlier detection from human wearable devices when used for Human Activity Recognition (HAR) and proposed a hybrid method combining the scores from several

outlier detection methods to improve HAR results. The main objective was to be able to optimally remove outliers in the dataset so that the data could better describe a set of underlying human activities. The study in [11] proposed an outlier detection method based on k-means that try to maximize both the quality and quantity of physical activity information captured by minimizing outliers.

The anomalies introduced in the data stream could be caused either by the internal functioning of the wearable sensor or by external sources (including the user wearing the device and environmental agents). When manufacturing wearable sensors for human monitoring, removing the data errors caused by the own device is very important [12]. The authors in [12] used an outlier filtering method based on statistically detecting outliers and specifying confidence levels using statistics. The algorithms used to perform outlier detection and filtering should be fast and low power consumption if they are to be implemented on wearable sensors. The authors in [13] proposed a fast and efficient method for outlier detection in data streams based on the way data points are selected from the data stream. However, the optimal removal of anomalies in human sensing remains a challenge in many scenarios [14].

In the line of using outliers in order to extract useful information for characterizing changes in the underlying data, some studies have recently proposed methods for detecting patterns and changes in user behavior that could require external action. The authors in [15] used activity recognition and abnormal behavior detection for elderly people with dementia in which activity recognition is considered as a sequence labelling problem, while abnormal behavior is flagged based on the deviation from normal patterns. The study in [16] used five different outlier detection methods on sensor data in order to produce contextual metrics in order to detect abnormal behavior for elderly people. The authors in [17] used outlier detection from pulse-rate, temperature, and pressure wearable sensors in order to enhance women safety. In the case proposed in [17], relevant events are translated into abnormal data points in which pulse-rate, temperature, and pressure move outside the normal range. Outlier detection methods are able to isolate and characterize them. In a different context, the authors in [18] used outlier detection techniques based on the Local Outlier Factor (LOF) method in order to analyze abnormal heart rate behavior while skiing. The authors in [9] proposed a method that uses outlier information from wearable sensors while performing a particular activity in order to detect sub-activities inside such main activities. The authors in [19] used outlier information from a GPS sensor while driving in order to detect different road elements such as traffic lights and street crossings. The detected outliers are characterized and classified using machine learning techniques providing good classification results while detecting road elements of interest. In a similar way, the main contribution in this paper is a new model that combines the use of detected outliers from inertial sensors and machine learning techniques in order to detect different obstacles which the user carrying the sensor device traverses while walking. This novel technique could be used to automatically create maps of elements such as stairs and slopes in cities by combining the data from different citizens.

## 3. Materials and Methods

The aim of this paper is to present an empirical study about a new procedure for using the information from multivariate outliers in mobile sensor data in order to detect different elements and physical barriers in an urban setting. Currently, thanks to the sensors embedded into mobile and wearable devices such as accelerometers, gyroscopes, barometers, magnetometers or GPS receivers, it is possible to gather a lot of user-related information in a non-intrusive, user-friendly way. To carry out this study, we have used the Android app Physics Toolbox Sensor Suite developed by Vieyra Software [20]. This application uses sensors embedded in mobile devices such as the linear accelerometer, gyroscope, inclinometer, barometer, proximity sensor, magnetometer, GPS sensor, light meter or color detector amongst others (even hygrometer and thermometer although these two sensors are available on a very limited number of smartphones). In addition, thanks to a multi-record option, the application allows to record data from multiple sensors at once.

Inside this section, the first sub-section presents the scenario chosen for this study and the second sub-section describes the data gathering process. Finally, in a third sub-section, the classification stage for multivariate outliers is presented.

### 3.1. Scenario

A new dataset has been gathered using a predefined route located in Madrid (Spain), corresponding to an open-air auditorium in the Enrique Tierno Galván Park, shown in Figure 1 (available at https://www.it.uc3m.es/~mario/Outliers/raw_data.rar). The same path has been walked four times in a row by different participants and each participant recording the data several times.

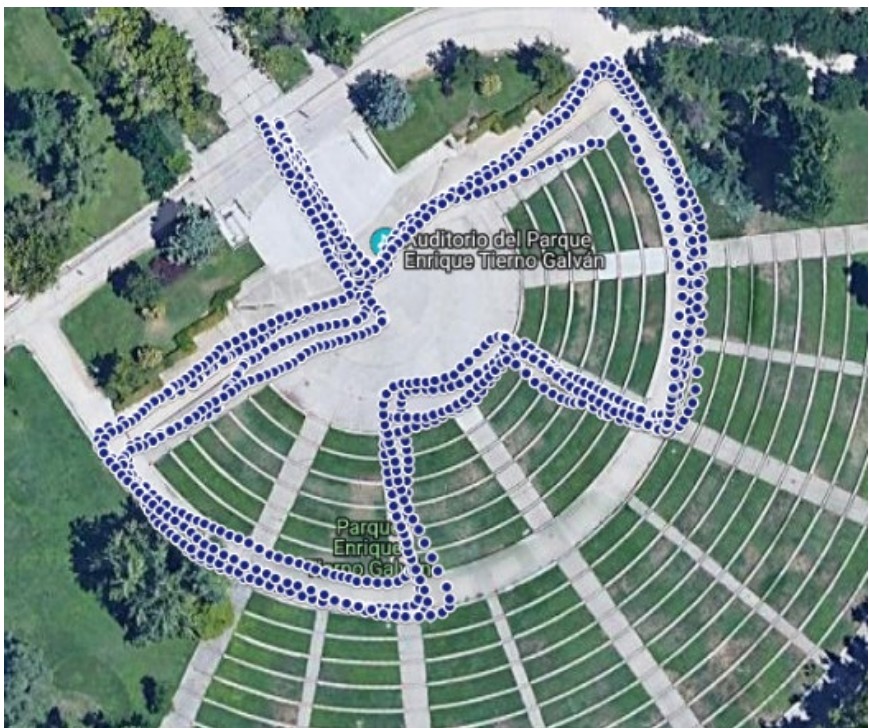

**Figure 1.** GPS traces along the path walked in the auditorium. (Image taken from Google Maps).

This scenario has been chosen for several reasons. First, we can find different situations/elements typically found while walking in urban areas such as walking up and down steps, turning right or left, and climbing up or down ramps/slopes. Second, there is a direct line of sight between the pedestrian who wears the mobile phone or device and the GPS satellites and therefore the GPS positioning errors are minimized. The GPS sensor will not be directly used for detecting urban elements/barriers but will tag the detected ones so that they can be validated in order to assess the precision and accuracy of the proposed method.

### 3.2. Data Gathering

The dataset has been collected from three different mobile devices as shown in Table 1. Moreover, the data recorded corresponds with three sensors: accelerometer, gyroscope, and GPS receiver. The accelerometer detects the orientation of the phone and measures the acceleration force in m/s$^2$ that is applied to the device with respect to *x*, y, and *z-axis*. Linear acceleration changes each time the mobile device accelerates, slows, or changes direction. On the other hand, the gyroscope measures rotational velocity in rad/s around the same device anchored three axes. In order to collect the data, several people have worn the mobile devices in the chest in a vertical position while walking the route four times without stopping. Each recording generated a new file in the dataset.

**Table 1.** Android smartphones used for gathering data.

| Mobile Device | Mean Sample Frequency | Number of Samples |
|---|---|---|
| bq Aquaris M5 | 90 Hz | 2,535,539 |
| Samsung Galaxy S4 | 235 Hz | 5,179,263 |
| Samsung Galaxy S7 | 149 Hz | 563,369 |

A multivariate variable made up of twelve different components is used to detect multivariate outliers. This multivariate variable is composed by processing the sampled sensor data each second. The components correspond to the mean and standard deviation values for the accelerometer in the *x-axis*, *y-axis* and *z-axis*, and the mean and standard deviation values for the gyroscope in the same three physical axes fragmenting the sensed data into one second windows. In addition, each observation is tagged with the GPS latitude and longitude coordinates in order to track the data samples. All missing data have been treated in the exploratory data analysis, removing some segments in the dataset or interpolating samples in the cases where it was possible.

The number of files recorded has been 52 that means 208 laps in the auditorium. The total number of samples that has been collected constitutes a total of 8,278,171, and corresponds to the three smartphones shown in Table 1. Moreover, Table 2 illustrates the number of observations or samples per second computed for each device with a total number of 53,742 observations of which 5778 have been detected as multivariate outliers.

**Table 2.** Multivariate outliers detected.

| Mobile Device | Sample/s | Number of Outliers | Percentage of Outliers |
|---|---|---|---|
| bq Aquaris M5 | 27,904 | 2780 | 9.96% |
| Samsung Galaxy S4 | 22,057 | 2609 | 12.10% |
| Samsung Galaxy S7 | 3781 | 389 | 10.27% |

In order to detect the multivariate outliers, we have used the algorithm described in [21], implemented with programming language *R*. There is no unique universal solution for the detection of multivariate outliers, and the best method depends on different parameters, such as the dimensions of the data structure, or the diverse kinds of data. It is worth noting, however, that a better solution for optimizing the results in multivariate outlier detection could be the combination of different types of techniques. The algorithm used in this research [21], developed by Peña y Prieto, combines the use of robust statistical methods and projections search techniques. It is based on projecting the data on the directions of maximum and minimum kurtosis since these projections have a high probability of showing outliers values if there are any. To be precise, any multivariate outlier should appear as an outlier in at least one projection direction, the one defined by the line that joins the data center with the outlier.

In univariate samples, a small proportion of outliers increases the kurtosis coefficient, which suggests investigating the directions where the projected points have maximum univariate kurtosis.

In addition, a large group of outliers can produce bimodality and low kurtosis, so it is also convenient to explore the directions where the projected points have minimal kurtosis.

The technique used is an iterative algorithm in which all observations suspected of being outliers are removed from the sample set in order to compute a robust means vector and a robust covariance matrix. With these robust estimators, the Mahalanobis distance [22] from the data centre to each point suspected of being an outlier is calculated, and the very distant points are considered outliers.

Once the final dataset has been transformed into a standardized sample, *Y*, with mean equal to zero and covariance matrix equal to the identity matrix, the algorithm consists of the following six basic steps:

1.  Calculate a set of $p$ orthogonal directions, $d_j$, that maximize the kurtosis coefficient of the data projected on these directions
2.  Obtain the projection $z_i$, of the observation $y_i$, on the direction $d_j$. Sample $Y$ is projected onto a space of dimension one unit less than $y_i$, which will be the sample in the next iteration
3.  Calculate the $p$ orthogonal directions, $d_j$, that minimize the kurtosis coefficient, until obtaining $2p$ directions
4.  Eliminate all samples suspected of being outliers in any one of the $2p$ directions, as those that meet a criterion experimentally compute
5.  Once the suspicious observations have been eliminated, the new robust vector of means and the robust covariance matrix are calculated
6.  All observations whose squared Mahalanobis distance between the sample and the robust vector of means is greater than a threshold, calculated as a function of $p$, are considered outliers

### 3.3. Classification

Once the outliers have been detected by the outlier detection algorithm in [21] each outlier is assigned to a particular area/region/zone based on the GPS coordinates. Each area corresponds to a homogeneous zone or physical barrier. The GPS sensor is embedded in the majority of smartphones providing the pedestrian location along the way corresponding to each multivariate outlier. In the selected scenario (previously described) twenty different areas have been identified in the chosen route, shown in Figure 2. These zones or areas are described in Table 3.

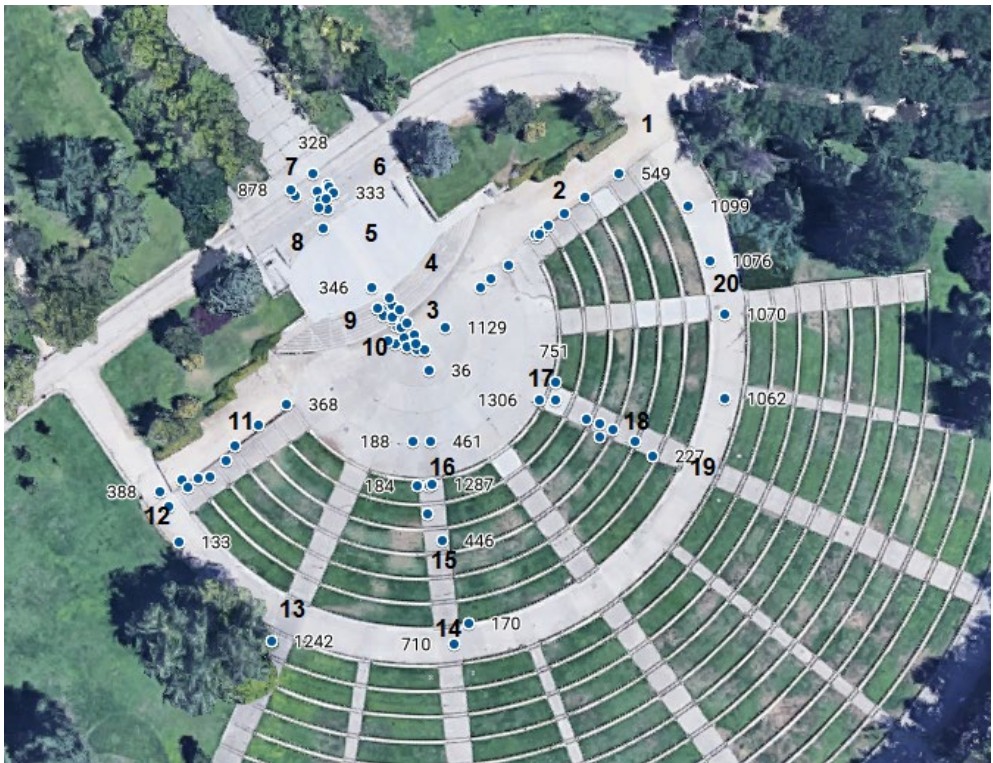

**Figure 2.** Location of some multivariate outliers in the different classes or zones. (Image taken from Google Maps).

**Table 3.** Number of multivariate outliers by each different zone and Mahalanobis distances (MD).

| Zone | Description | Number of Outliers | Max. MD | Min. MD |
|------|-------------|--------------------|---------|---------|
| 1 | Left turn with descending ramp | 258 | 280.06 | 27.17 |
| 2 | Descending ramp | 39 | 900.44 | 28.38 |
| 3 | Right turn with rising high step | 278 | 5796.72 | 28.85 |
| 4 | Rising high stairs | 915 | 2996.37 | 26.91 |
| 5 | Straight line | 27 | 84.56 | 27.27 |
| 6 | Rising stairs | 182 | 749.41 | 29.55 |
| 7 | 180-degree turn | 532 | 1130.52 | 27.22 |
| 8 | Stair descent | 118 | 167.04 | 27.40 |
| 9 | Descending high stairs | 625 | 395.92 | 26.74 |
| 10 | Descent step with right turn | 137 | 404.50 | 28.32 |
| 11 | Ascent ramp | 145 | 165.34 | 26.76 |
| 12 | Final ascent ramp with left turn | 383 | 1498.24 | 29.43 |
| 13 | Semi-circular path | 22 | 1616.70 | 36.60 |
| 14 | Left turn with descending step | 399 | 238.83 | 26.92 |
| 15 | Descending ramp with steps | 347 | 236.22 | 26.86 |
| 16 | Right turn with descending step | 340 | 345.26 | 27.55 |
| 17 | Right turn with rising step | 380 | 258.11 | 27.74 |
| 18 | Rising ramp with steps | 293 | 248.96 | 26.86 |
| 19 | Left turn with rising step | 325 | 298.52 | 30.83 |
| 20 | Semi-circular path | 18 | 254.83 | 34.10 |

Figure 2 also shows some of the observations detected as multivariate outliers, along with the corresponding observation number in the dataset.

An additional zero zone is identified in the data analysis which corresponds to the beginning and the end of the route where the pedestrian is standing without moving, but outliers values located in this zone have not been taken into account. With this information, the 5778 remaining outliers detected have been labeled according to the different zones they belong to. Each zone has been modelled as a parallelogram defined by the GPS coordinates of its corners.

Table 3 illustrates the total number of outliers that appear in each zone and the maximum and the minimum Mahalanobis distance found in these outliers. These outliers are vectors of twelve features, corresponding to the mean and standard deviation of the accelerometer and gyroscope measures in the three Cartesian coordinates. Therefore, the outliers will be used to classify a determinate zone, since they must have representative values and significant data to identify one zone or another. Therefore, it does not matter what the cause of these outliers is, or what they represent, but the fact that they can identify a certain area.

We can see on the one hand, that the zone where a higher number of outliers appears is the zone 4 with a total number of 915 multivariate outliers. That is not surprising since this zone corresponds to a going upstairs area whose steps have a double height compared to the mean height for normal steps (that it is typically about 18 cm). The maximum Mahalanobis distance corresponds to zone 3. This can be explained by the fact that this zone combines a step with right turn. On the other hand, the zones with a fewer number of outliers are zone 20, zone 13 and zone 5. This finding is also not surprising since in these zones there are not significant movements that deviate from normal walking. On the contrary, the next two zones with more outlier values are zone 9 and zone 7.

The zone labelled as 9, again, corresponds with the stairs with double height, in this case, for a descending walk instead of walking up. Finally, zone 7 corresponds with a 180-degree turn, which it is a different situation with respect to the rest of the zones.

In order to classify the different zones or classes based on the multivariate outliers, three different features have been extracted. First, the number of outlier values located in every zone. Second, the average of the norm for the accelerometer components. Finally, the average of the norm for the gyroscope components.

Among the different supervised classification approaches, we have used the technique of Support Vector Machines or SVM with radial basis function (RBF) kernel [23] since SVM offers better accuracy compared to other shallow learning-based classifiers and it is one of the most efficient algorithm in data mining [24]. The experimental studies reported in the literature have been mainly aimed to compare different families of classifiers [25]. Regarding the SVM, it cannot be said that there is a kernel that outperforms the rest, it depends to a great extent on the nature of the problem that is being treated. However, it is highly recommended to test the RBF kernel because this kernel only has two hyperparameters to optimize, the gamma parameter and the C penalty, common to all SVM approaches [26].

Other kernels have been used with the worst result, as Table 4 shows, so the radial kernel has been chosen. We have not used deep learning-based approaches since they require a bigger dataset than the one generated for this study. All the results obtained are presented below, in the next section.

**Table 4.** Accuracy Results for different SVM kernels.

| Kernel | Accuracy Rate | Kappa Coefficient | Training Time |
|---|---|---|---|
| Linear | 73.48% | 0.72 | 1.62 s |
| Radial | 74.90% | 0.73 | 3.76 s |
| Polynomial | 57.12% | 0.54 | 1.62 s |
| Sigmoid | 34.13% | 0.30 | 1.79 s |

## 4. Results and Discussion

The sensor data in the 52 repetitions of the route, each repetition containing 4 laps of the path shown in Figure 1, has been used in order to validate the results of the proposed method. For each repetition of the route, and for each zone, the number of outlier values detected in each zone, the average of the norm for the accelerometer components and the average of the norm for the gyroscope components have been computed as described in the previous section.

The first result obtained refers to a multiclass classification. The twenty zones or classes have been classified using a 10-fold cross-validation schema for all the three features previously described (the number of outlier values detected in each zone, the accelerometer norm and the gyroscope norm average values). For these twenty classes or zones, the data for training and test correspond to a total number of 1040 observations. Table 5 illustrates the confusion matrix for the SVM classifier, that is, the table of hits and errors, where we can see if the predicted values match the actual values [27]. The result shows that the average accuracy rate for all classes is 74.90% with a kappa coefficient of 0.73. This result is the best compared with other kernels as Table 4 shows. Kappa coefficient or global accuracy index is a measure of the difference between the accuracy achieved with an automatic classifier and the probability of achieving a correct classification with a random classifier, with a maximum value of 1.0 representing an accuracy of 100%. Moreover, Table 6 shows the statistics by each class.

The confusion matrix shows interesting results. First, zones 1, 4, 7, 8, 9 and 15 have 100% of specificity or precision. In addition, zones 12, 13, 14 and 19 have 99% of specificity. The results of this research are not surprising if we consider the specific scenario and the characteristics of the situations under study.

The results using a 10-fold cross-validation technique for all the three features presented in Tables 5 and 6, with a radial kernel SVM classifier is about of 75% accuracy for all the twenty zones when they are considered as different elements/barriers which may have particular characteristics (this is a worst case scenario since some of the zones are very similar in their physical structure as can be seen in Figure 2, and represent therefore the same kind of element/barrier). The best accuracy result is for zone 7, which corresponds to a 180-degree turn and achieves a 100% prediction. On the contrary, the worst result is for zone 6 with a 50% of sensitivity, followed by zone 18 with about 52% of sensitivity. Both zones correspond with going upstairs of a usual height, zone 18 also capturing a partial going up a ramp segment.

**Table 5.** Confusion matrix using the SVM classifier with RBF kernel.

| Ref./Pred. | 1 | 2 | 3 | 4 | 5 | 6 | 7 | 8 | 9 | 10 | 11 | 12 | 13 | 14 | 15 | 16 | 17 | 18 | 19 | 20 |
|---|---|---|---|---|---|---|---|---|---|---|---|---|---|---|---|---|---|---|---|---|
| 1 | 34 | 0 | 0 | 0 | 0 | 0 | 0 | 0 | 0 | 0 | 0 | 0 | 0 | 0 | 0 | 0 | 0 | 0 | 0 | 0 |
| 2 | 0 | 50 | 0 | 2 | 8 | 7 | 0 | 5 | 1 | 0 | 0 | 0 | 0 | 0 | 1 | 0 | 0 | 0 | 0 | 0 |
| 3 | 7 | 0 | 43 | 1 | 0 | 4 | 0 | 0 | 0 | 0 | 0 | 3 | 0 | 4 | 0 | 3 | 0 | 0 | 8 | 0 |
| 4 | 0 | 0 | 0 | 32 | 0 | 0 | 0 | 0 | 0 | 0 | 0 | 0 | 0 | 0 | 0 | 0 | 0 | 0 | 0 | 0 |
| 5 | 0 | 0 | 0 | 0 | 43 | 0 | 0 | 6 | 0 | 0 | 0 | 0 | 6 | 0 | 2 | 0 | 0 | 0 | 0 | 6 |
| 6 | 0 | 0 | 1 | 7 | 1 | 26 | 0 | 0 | 5 | 0 | 18 | 0 | 1 | 0 | 0 | 0 | 0 | 15 | 0 | 1 |
| 7 | 0 | 0 | 0 | 0 | 0 | 0 | 52 | 0 | 0 | 0 | 0 | 0 | 0 | 0 | 0 | 0 | 0 | 0 | 0 | 0 |
| 8 | 0 | 0 | 0 | 0 | 0 | 0 | 0 | 38 | 0 | 0 | 0 | 0 | 0 | 0 | 0 | 0 | 0 | 0 | 0 | 0 |
| 9 | 0 | 0 | 0 | 0 | 0 | 0 | 0 | 0 | 42 | 0 | 0 | 0 | 0 | 0 | 0 | 0 | 0 | 0 | 0 | 0 |
| 10 | 10 | 0 | 0 | 1 | 0 | 0 | 0 | 0 | 0 | 52 | 0 | 2 | 0 | 2 | 0 | 1 | 2 | 1 | 10 | 0 |
| 11 | 0 | 0 | 0 | 1 | 0 | 0 | 0 | 0 | 2 | 0 | 23 | 0 | 0 | 0 | 3 | 0 | 0 | 5 | 0 | 0 |
| 12 | 0 | 0 | 0 | 0 | 0 | 0 | 0 | 0 | 0 | 0 | 1 | 32 | 0 | 1 | 0 | 0 | 0 | 0 | 0 | 0 |
| 13 | 1 | 0 | 0 | 0 | 0 | 1 | 0 | 0 | 0 | 0 | 0 | 0 | 45 | 1 | 0 | 0 | 1 | 0 | 1 | 0 |
| 14 | 0 | 0 | 0 | 0 | 0 | 0 | 0 | 0 | 0 | 0 | 0 | 0 | 0 | 31 | 0 | 0 | 2 | 0 | 0 | 0 |
| 15 | 0 | 0 | 0 | 0 | 0 | 0 | 0 | 0 | 0 | 0 | 0 | 0 | 0 | 0 | 46 | 0 | 0 | 0 | 0 | 0 |
| 16 | 0 | 0 | 2 | 0 | 0 | 4 | 0 | 0 | 0 | 0 | 0 | 10 | 0 | 1 | 0 | 47 | 6 | 4 | 1 | 0 |
| 17 | 0 | 0 | 0 | 0 | 0 | 0 | 0 | 0 | 0 | 0 | 0 | 5 | 0 | 11 | 0 | 0 | 39 | 0 | 0 | 0 |
| 18 | 0 | 0 | 0 | 8 | 0 | 8 | 0 | 0 | 0 | 0 | 10 | 0 | 0 | 0 | 0 | 0 | 0 | 27 | 0 | 0 |
| 19 | 0 | 0 | 6 | 0 | 0 | 0 | 0 | 0 | 0 | 0 | 0 | 0 | 0 | 0 | 0 | 0 | 2 | 0 | 32 | 0 |
| 20 | 0 | 2 | 0 | 0 | 0 | 2 | 0 | 3 | 2 | 0 | 0 | 0 | 0 | 1 | 0 | 1 | 0 | 0 | 0 | 45 |

**Table 6.** Statistics by class.

| Class | Sensitivity | Specificity | Positive Pred. Value | Negative Pred. Value | Balanced Accuracy |
|---|---|---|---|---|---|
| 1 | 0.65385 | 1.00000 | 1.00000 | 0.98211 | 0.82692 |
| 2 | 0.96154 | 0.97571 | 0.67568 | 0.99793 | 0.96862 |
| 3 | 0.82692 | 0.96964 | 0.58904 | 0.99069 | 0.89828 |
| 4 | 0.61538 | 1.00000 | 1.00000 | 0.98016 | 0.80769 |
| 5 | 0.82692 | 0.97976 | 0.68254 | 0.99079 | 0.90334 |
| 6 | 0.50000 | 0.95040 | 0.34667 | 0.97306 | 0.72520 |
| 7 | 1.00000 | 1.00000 | 1.00000 | 1.00000 | 1.00000 |
| 8 | 0.73077 | 1.00000 | 1.00000 | 0.98603 | 0.86538 |
| 9 | 0.80769 | 1.00000 | 1.00000 | 0.98998 | 0.90385 |
| 10 | 1.00000 | 0.97065 | 0.64198 | 1.00000 | 0.98532 |
| 11 | 0.44231 | 0.98887 | 0.67647 | 0.97117 | 0.71559 |
| 12 | 0.61538 | 0.99798 | 0.94118 | 0.98012 | 0.80668 |
| 13 | 0.86538 | 0.99494 | 0.90000 | 0.99293 | 0.93016 |
| 14 | 0.59615 | 0.99798 | 0.93939 | 0.97915 | 0.79706 |
| 15 | 0.88462 | 1.00000 | 1.00000 | 0.99396 | 0.94231 |
| 16 | 0.90385 | 0.97166 | 0.62667 | 0.99482 | 0.93775 |
| 17 | 0.75000 | 0.98381 | 0.70909 | 0.98680 | 0.86690 |
| 18 | 0.51923 | 0.97368 | 0.50943 | 0.97467 | 0.74646 |
| 19 | 0.61538 | 0.99190 | 0.80000 | 0.98000 | 0.80364 |
| 20 | 0.86538 | 0.98887 | 0.80357 | 0.99289 | 0.92713 |

Regarding zone 6, there are 18 observations that are considered as zone 11, and 15 observations misclassified as zone 18. All these 3 regions are very similar in structure based on going upstairs of similar dimensions and therefore they could be merged together in the same class of physical barrier. The same happens for zone 18 in which we find observations classified as the other going upstair/ramp zones or segments. Regarding zones 13 and 20, it should be expected that no outlier appeared, due to normal walking conditions, as there is nothing significant found while traversing them. However, these zones show some residual outliers that would correspond to various situations. This can be explained by the fact that in zone 13 there is a sewer cover. Also, as a consequence of crossing with

dogs and people, or by dodging water jets from sprinklers. Furthermore, several outliers may appear as a result of sneezing, due to allergies to pollen since the mobile device is located right on the chest. Regarding zone 5, the most likely reason for the manifestation of outliers is due to labelling errors (border points in which the GPS errors could misallocate some of the samples).

As a next part of the study, we evaluated zones 1, 12, 14, 16, 17 and 19 together. Zones 1, 12, 14 and 19 correspond to a turning to the left. Besides, zone 1 starts going down a ramp and zone 12 starts in the end of going up a ramp. Furthermore, zone 14 is the beginning of down a step and zone 19 starts with the ending of going up a step. Finally, zones 16 and 17 correspond to a turning to the right, zone 16 to the ending of going down a step and zone 17 to the starting of going up a step.

Table 7 shows the confusion matrix of these six classes with an average accuracy of 91.35% and a coefficient kappa of 0.89. There are five situations of zone 1 predicted as zone 14, which it is not surprising since both classes are very similar. And, there are also four situations of zone 17 predicted as zone 12, which it seems not to make much sense. Table 8 illustrates the statistics by these classes.

**Table 7.** Confusion Matrix using the SVM classifier in zones 1, 12, 14, 16, 17 and 19.

| Reference/ Prediction | 1 | 12 | 14 | 16 | 17 | 19 | Average Accuracy | Kappa Coefficient |
|---|---|---|---|---|---|---|---|---|
| 1 | 47 | 3 | 3 | 1 | 0 | 2 | | |
| 12 | 0 | 48 | 1 | 1 | 4 | 0 | | |
| 14 | 5 | 1 | 47 | 0 | 1 | 0 | 91.35% | 0.8962 |
| 16 | 0 | 0 | 1 | 49 | 2 | 1 | | |
| 17 | 0 | 0 | 0 | 0 | 45 | 0 | | |
| 19 | 0 | 0 | 0 | 1 | 0 | 49 | | |

**Table 8.** Statistics by class for zones 1, 12, 14, 16, 17 and 19.

| Class | Sensitivity | Specificity | Positive Pred. Value | Negative Pred. Value | Balanced Accuracy |
|---|---|---|---|---|---|
| 1 | 0.9038 | 0.9654 | 0.8393 | 0.9805 | 0.9346 |
| 12 | 0.9231 | 0.9769 | 0.8889 | 0.9845 | 0.9500 |
| 14 | 0.9038 | 0.9731 | 0.8704 | 0.9806 | 0.9385 |
| 16 | 0.9423 | 0.9846 | 0.9245 | 0.9884 | 0.9635 |
| 17 | 0.8654 | 1.0000 | 1.0000 | 0.9738 | 0.9327 |
| 19 | 0.9423 | 0.9962 | 0.9800 | 0.9885 | 0.9692 |

In addition, Table 9 shows the confusion matrix of zones 2, 11, 15, and 18, and Table 10 the statistics by class. These zones are very similar. On the one hand, zones 2 and 15 correspond both with descending a ramp. On the other hand, zones 11 and 18 correspond with going up a ramp, but zones 15 and 18 have also steps. The average accuracy is 94.7% and the coefficient kappa is 0.9. There are six predictions of zone 11 that correspond with zone 18, which is quite reasonable.

**Table 9.** Confusion Matrix using the SVM classifier in zones 2, 11, 15 and 18.

| Reference/ Prediction | 2 | 11 | 15 | 18 | Average Accuracy | Kappa Coefficient |
|---|---|---|---|---|---|---|
| 2 | 52 | 0 | 2 | 0 | | |
| 11 | 0 | 51 | 2 | 6 | 94.71% | 0.9295 |
| 15 | 0 | 0 | 48 | 0 | | |
| 18 | 0 | 1 | 0 | 46 | | |

**Table 10.** Statistics by class for zones 2, 11, 15 and 18.

| Class | Sensitivity | Specificity | Positive Pred. Value | Negative Pred. Value | Balanced Accuracy |
|-------|-------------|-------------|----------------------|----------------------|-------------------|
| 2 | 1.0000 | 0.9872 | 0.9630 | 1.0000 | 0.9936 |
| 11 | 0.9808 | 0.9487 | 0.8644 | 0.9933 | 0.9647 |
| 15 | 0.9231 | 1.0000 | 1.0000 | 0.9750 | 0.9615 |
| 18 | 0.8846 | 0.9936 | 0.9787 | 0.9627 | 0.9391 |

Finally, zones 3, 4, 9, and 10 have been classified. Tables 11 and 12 show the confusion matrix and the statistics by class, respectively.

**Table 11.** Confusion Matrix using the SVM classifier in zones 3, 4, 9 and 10.

| Reference/ Prediction | 3 | 4 | 9 | 10 | Average Accuracy | Kappa Coefficient |
|-----------------------|-----|-----|-----|-----|------------------|-------------------|
| 3 | 52 | 2 | 0 | 0 | | |
| 4 | 0 | 45 | 2 | 0 | 95.67% | 0.9423 |
| 9 | 0 | 4 | 50 | 0 | | |
| 10 | 0 | 1 | 0 | 52 | | |

**Table 12.** Statistics by class for zones 3, 4, 9 and 10.

| Class | Sensitivity | Specificity | Positive Pred. Value | Negative Pred. Value | Balanced Accuracy |
|-------|-------------|-------------|----------------------|----------------------|-------------------|
| 3 | 1.0000 | 0.9872 | 0.9630 | 1.0000 | 0.9936 |
| 4 | 0.8654 | 0.9872 | 0.9574 | 0.9565 | 0.9263 |
| 9 | 0.9615 | 0.9744 | 0.9259 | 0.9870 | 0.9679 |
| 10 | 1.0000 | 0.9936 | 0.9811 | 1.0000 | 0.9968 |

Zones 3 and 10 correspond with turning to the right, and zones 4 and 9 with going up and down double steps, respectively. However, zones 3 and 10 can be easily mislabeled because of the lack of precision regarding the GPS coordinates provided by the mobile device. Still, the average accuracy is about 95.7% and the coefficient kappa is 0.94.

## 5. Conclusions

The aim of this paper has been to present a new use for multivariate outliers. The dataset has been gathered from sensors embedded in mobile devices while walking in an urban scenario located in a park, in order to analyze and process these data, together with the application of statistical and machine learning techniques, and with the objective of identifying anomalous situations while walking.

The experimental results have been calculated with an SVM algorithm with a radial basis function kernel. Regarding all the twenty classes, the algorithm is able to classify half of the zones with specificity values above 0.99 and only getting significant confusion figures among zones which correspond to similar physical elements such as going upstairs/ramps. When used with fewer classes, for instance, four, the accuracy rate of the proposed algorithm is about 95%.

The results have been obtained using just three features and only one kernel. For future work, a more exhaustive analysis of other features describing the sensor data, as well as other kernel functions will be analyzed in order to increase the achieved accuracy.

Finally, some limitations of this study should be noted and discussed. First, there are some GPS errors that would result in mislabeled zones and second, errors generated by the use of different mobile devices that resulted in changes in the sampling frequencies/rates which required the use of data interpolation techniques for each of the smartphones.

Despite the above limitations, the research findings demonstrate that it is a novel approach to the use of outlier detection, as a mechanism for automatic detection of anomalous situations while walking, although it could be applied in any other kind of situations.

In summary, the most significant conclusion of this study has been that outlier values can be very useful to identify different situations and represent valuable features not used in other research. Future works should collect data using a larger number of different mobile devices and to gather the greatest amount of data possible.

**Author Contributions:** R.R.B. performed the data gathering and preprocessing, contributed to the development of the data processing algorithms, executed the detailed analysis and led the writing of the document. M.M.-O. set the objectives of the research, contributed to the design of the algorithms, guided the data acquisition, validated the data processing and results, designed the paper, wrote some sections and performed the final corrections. All authors have read and agreed to the published version of the manuscript.

**Funding:** The research leading to these results has received funding from the "ANALYTICS USING SENSOR DATA FOR FLATCITY" project TIN2016-77158-C4-1-R (MINECO/ERDF, EU) funded by the Spanish Agencia Estatal de Investigación (AEI) and the European Regional Development Fund (ERDF). Furthermore, the first author is supported by the MINECO Grant nº: BES-2014-070462

**Acknowledgments:** The first author wishes to thank her colleagues Rosa Matamoros and Antonio Herrera for their help to collect data.

**Conflicts of Interest:** The authors declare no conflict of interest. The funders had no role in the design of the study; in the collection, analyses, or interpretation of data; in the writing of the manuscript, or in the decision to publish the results.

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
