# Peer review of "Using Multivariate Outliers from Smartphone Sensor Data to Detect Physical Barriers While Walking in Urban Areas"

_technologies, doi:10.3390/technologies8040058_

Round 1

Reviewer 1 Report

Paper deals with very important task for today. Authors used SVM for outliers detection in sensor data. 

1. Lines 48-50 - please add the references for the describing methods.

2. Related works section is absent. Authors Should add it to the paper. It should contain the overview and analysis of the up-to-day SVM methods for solving the classification task. Authors should use this accurately SVM-modification (DOI: 10.5815/ijisa.2018.09.05) among many others.

3. In the paper I cant find any information about omnission in data. IoT-based data always have omission and anomalies for different reasons that was investigated there - DOI: 10.3390/s20092625  or DOI : 10.2174/2210327910999200813151904

4. It would be good to see the links in the open access repository for the dataset used for modeling.

5. Lines 165-170. Authors should argue they choose on the SVM kernel. Why rbf kernel? Did authors try other kernels like there - (DOI: 10.5815/ijisa.2018.09.05). It would be good to see the results of such study.

6. It would be good to see the training time for the proposed method. The simplest comparison is to compare the performance of SVM using different kernels. However, it would also be good to compare with other machine learning algorithms.

7. Authors did not provide the comparison with other ML-based methods. It should be done. The simplest way to compare is to compare the performance of SVM using different kernels. However, it would be good to see a comparison with other methods of machine learning.

8. Most of the references are 10 and more years old. How will the authors argue the actuality of this task? Authors should add up-to-date references. Maybe it would be done when authors fix my suggestion #2.  

9. However, the biggest problem is the scientific novelty of the work. Authors should clearly formulate it and submit it at the end of the omitted section (Related works). Otherwise, I will have to reject this paper.

Author Response

  1. Lines 48-50 - please add the references for the describing methods.

                Done.

  1. Related works section is absent. Authors Should add it to the paper. It should contain the overview and analysis of the up-to-day SVM methods for solving the classification task. Authors should use this accurately SVM-modification (DOI: 10.5815/ijisa.2018.09.05) among many others.

                Done in lines 58-71 and 220-225.

  1. In the paper I can’t find any information about omnission in data. IoT-based data always have omission and anomalies for different reasons that was investigated there –

DOI: 10.3390/s20092625 or DOI : 10.2174/2210327910999200813151904

                Done in lines 129-130.

  1. It would be good to see the links in the open access repository for the dataset used for modeling.

            The entire dataset has been published open access ant the reference to it has been added to line 102.

  1. Lines 165-170. Authors should argue they choose on the SVM kernel. Why rbf kernel? Did authors try other kernels like there - (DOI: 10.5815/ijisa.2018.09.05). It would be good to see the results of such study.

            Done in Table 5.

  1. It would be good to see the training time for the proposed method. The simplest comparison is to compare the performance of SVM using different kernels. However, it would also be good to compare with other machine learning algorithms.

The training times for different kernels have also been added to Table 5.

  1. Authors did not provide the comparison with other ML-based methods. It should be done. The simplest way to compare is to compare the performance of SVM using different kernels. However, it would be good to see a comparison with other methods of machine learning.

The performance for different kernels is shown. For future studies, other classifiers will also be added.

  1. Most of the references are 10 and more years old. How will the authors argue the actuality of this task? Authors should add up-to-date references. Maybe it would be done when authors fix my suggestion #2.

References 8 and 9 are now showing newer related studies

  1. However, the biggest problem is the scientific novelty of the work. Authors should clearly formulate it and submit it at the end of the omitted section (Related works). Otherwise, I will have to reject this paper.

            Done in lines 75-80.

Reviewer 2 Report

The study is well done, and the paper is well written. 

But a few points will improve the paper: 

  1. The introduction could be made a bit more clearer. 
  2. The outlier needs to be defined. For example, when the authors say that zone 2 has 39 outliers, what does that mean? Are people deviating from the normal path? What exactly is defining the outlier?
  3. Also, what is the purpose of detecting the outliers -- this has not been made clear.
  4. The features used in this study have not been made very clear. In section 2.2 it is stated that a multivariate variable made up to twelve different components is used to detect multivariate outliers. It goes on to say, "The components correspond to the mean and standard deviation values ... in the x-axis, y-axis, ... I was assuming that these were being used as the features.
  5. The thrust of this paper is in determining the outliers, so the outlier detection algorithm should be elaborated on a little more. It is very briefly explained in section 2.3.
  6. In Figure 2, there are some numbers besides the zone numbers - what are they? For example, what is the 388 beside the 12?
  7. Though the techniques are known techniques, for the sake of completeness, they should be at least brief explained -- Mahalanobis distance and SVM. Why is the RBF kernel used -- this should be briefly mentioned.
  8. New references should be used. There is newer work on outlier detection.
  9. A "Related Work" section would enhance the paper. Papers related to accelerometer/gyroscope data and outlier detection should be referenced.

Author Response

  1. The introduction could be made a bit more clearer.

                Done in lines 52-80.

  1. The outlier needs to be defined. For example, when the authors say that zone 2 has 39 outliers, what does that mean? Are people deviating from the normal path? What exactly is defining the outlier?

            Done in lines 198-199.

  1. Also, what is the purpose of detecting the outliers -- this has not been made clear.

                Done in lines 72-80.

  1. The features used in this study have not been made very clear. In section 2.2 it is stated that a multivariate variable made up to twelve different components is used to detect multivariate outliers. It goes on to say, "The components correspond to the mean and standard deviation values ... in the x-axis, y-axis, … I was assuming that these were being used as the features.

                Done in lines 194-196.

  1. The thrust of this paper is in determining the outliers, so the outlier detection algorithm should be elaborated on a little more. It is very briefly explained in section 2.3.

            Done in lines 147-172.

  1. In Figure 2, there are some numbers besides the zone numbers - what are they? For example, what is the 388 beside the 12?

                Done in lines 186-187.

  1. Though the techniques are known techniques, for the sake of completeness, they should be at least brief explained -- Mahalanobis distance and SVM. Why is the RBF kernel used -- this should be briefly mentioned.

            Done in lines 221-227.

  1. New references should be used. There is newer work on outlier detection.

References 8 and 9 are now showing newer related studies

  1. A "Related Work" section would enhance the paper. Papers related to accelerometer/gyroscope data and outlier detection should be referenced.

The related work section was merged with the introduction section but we could divide the information into 2 sections if preferred.

Round 2

Reviewer 1 Report

The idea of the paper is good. The authors try to improve the paper but:

  1. In general, the article is well written. Most of the necessary formal things are present. However, only formal. There are no a sufficient level of scientific novelty in this paper!
  2. There are no Related works section in this paper. The authors added two sources, but this is not enough. The authors had to spend more time and carry out a systematic review of the literature in scientometric databases. And the sentence in 68-71 does not justify the situation at all. Its like a conference paper!
  3. Most of the references are 20 years ago. The authors did not prove the importance of their research!

Based on these, I cannot recommend this paper for publication in such a recognized Journal. I think this is a conference paper and it should be published in some conference.

Author Response

Reviewer 1

The idea of the paper is good. The authors try to improve the paper but:

  1. In general, the article is well written. Most of the necessary formal things are present. However, only formal. There are no a sufficient level of scientific novelty in this paper!

Answer:

The novelty of the paper has been better justified inside the paper. A proper review of the related work has been added and the novelty of the paper has been better motivated. The following text has been added to the paper:

The main contribution in this paper is a new model that combines the use of detected outliers from inertial sensors and machine learning techniques in order to detect different obstacles which the user carrying the sensor device traverses while walking. This novel technique could be used to automatically create maps of elements such as stairs and slopes in cities by combining the data from different citizens

  1. There are no Related works section in this paper. The authors added two sources, but this is not enough. The authors had to spend more time and carry out a systematic review of the literature in scientometric databases. And the sentence in 68-71 does not justify the situation at all. Its like a conference paper!

Answer:

An entire new section for the related work has been added:

  1. Related work

Data generated by wearable sensors are prone to different types of anomalies. Price sensitive, low cost and battery powered wearable devices provide in many cases data streams in which sensed data is mixed with outliers of various kinds. In some cases, outlier detection is needed in order to remove random errors introduced in the data and therefore purify the data quality in order to better learn the intricate patterns in the data [10]. In other cases, outlier detection will be a valuable source of information in order to detect changes in behavior or in the underlying characteristics of the data [10].

In the line of removing outliers from wearable sensors, the authors in [10] provided a review of different methods that have been used for outlier detection from human wearable devices when used for Human Activity Recognition (HAR) and proposed a hybrid method combining the scores from several outlier detection methods to improve HAR results. The main objective was to be able to optimally remove outliers in the dataset so that the data could better describe a set of underlying human activities. The study in [11] proposed an outlier detection method based on k-means that try to maximize both the quality and quantity of physical activity information captured by minimizing outliers.

The anomalies introduced in the data stream could be caused either by the internal functioning of the wearable sensor or by external sources (including the user wearing the device and environmental agents). When manufacturing wearable sensors for human monitoring, removing the data errors caused by the own device is very important [12]. The authors in [12] used an outlier filtering method based on statistically detecting outliers and specifying confidence levels using statistics. The algorithms used to perform outlier detection and filtering should be fast and low power consumption if they are to be implemented on wearable sensors. The authors in [13] proposed a fast and efficient method for outlier detection in data streams based on the way data points are selected from the data stream. However, the optimal removal of anomalies in human sensing remains a challenge in many scenarios [14].

In the line of using outliers in order to extract useful information for characterizing changes in the underlying data, some studies have recently proposed methods for detecting patterns and changes in user behavior that could require external action. The authors in [15] used activity recognition and abnormal behavior detection for elderly people with dementia in which activity recognition is considered as a sequence labelling problem, while abnormal behavior is flagged based on the deviation from normal patterns. The study in [16] used five different outlier detection methods on sensor data in order to produce contextual metrics in order to detect abnormal behavior for elderly people. The authors in [17] used outlier detection from pulse-rate, temperature and pressure wearable sensors in order to enhance women safety. In the case proposed in [17], relevant events are translated into abnormal data points in which pulse-rate, temperature and pressure move outside the normal range. Outlier detection methods are able to isolate and characterize them. In a different context, the authors in [|8] used outlier detection techniques based on the Local Outlier Factor (LOF) method in order to analyze abnormal heart rate behavior while skiing. The authors in [9] proposed a method that uses outlier information from wearable sensors while performing a particular activity in order to detect sub-activities inside such main activities. The authors in [19] used outlier information from a GPS sensor while driving in order to detect different road elements such as traffic lights and street crossings. The detected outliers are characterized and classified using machine learning techniques providing good classification results while detecting road elements of interest. In a similar way, the main contribution in this paper is a new model that combines the use of detected outliers from inertial sensors and machine learning techniques in order to detect different obstacles which the user carrying the sensor device traverses while walking. This novel technique could be used to automatically create maps of elements such as stairs and slopes in cities by combining the data from different citizens.     

  1. Most of the references are 20 years ago. The authors did not prove the importance of their research!

Answer:

Ten new references have been added:

  1. Gopalakrishnan, N., & Krishnan, V. (2019). Improving data classification accuracy in sensor networks using hybrid outlier detection in HAR. Journal of Intelligent & Fuzzy Systems, 37(1), 771-782.
  2. Jones, P. J., James, M. K., Davies, M. J., Khunti, K., Catt, M., Yates, T., ... & Mirkes, E. M. (2020). FilterK: A new outlier detection method for k-means clustering of physical activity. Journal of Biomedical Informatics, 103397.
  3. Vafeas, A. T., Fafoutis, X., Elsts, A., Craddock, I. J., Biswas, M. I., Piechocki, R. J., & Oikonomou, G. (2020, February). Wearable Devices for Digital Health: The SPHERE Wearable 3. In Embedded Wireless Systems and Networks (EWSN): On-Body Sensor Networks (OBSN 2020).
  4. Yoon, S., Lee, J. G., & Lee, B. S. (2019). NETS: extremely fast outlier detection from a data stream via set-based processing. Proceedings of the VLDB Endowment, 12(11), 1303-1315.
  5. Liu, X., Chen, H., Montieri, A., & Pescapè, A. (2020). Human behavior sensing: challenges and approaches. Journal of Ambient Intelligence and Humanized Computing, 1-16.
  6. Arifoglu and A. Bouchachia, Activity Recognition and Abnormal Behaviour Detection with Recurrent Neural Networks, The 14th International Conference on Mobile Systems and Pervasive Computing (MobiSPC 2017) (2017), 86–93.
  7. Koutli, M., Theologou, N., Tryferidis, A., & Tzovaras, D. (2019, October). Abnormal Behavior Detection for elderly people living alone leveraging IoT sensors. In 2019 IEEE 19th International Conference on Bioinformatics and Bioengineering (BIBE) (pp. 922-926). IEEE.
  8. Hyndavi, V., Nikhita, N. S., & Rakesh, S. (2020, June). Smart Wearable Device for Women Safety Using IoT. In 2020 5th International Conference on Communication and Electronics Systems (ICCES) (pp. 459-463). IEEE.
  9. Yue, N., & Claes, S. Wearable Sensors for Smart Abnormal Heart Rate Detection during Skiing. Internet Technology Letters, e230.
  10. Munoz-Organero, M., Ruiz-Blaquez, R., & Sánchez-Fernández, L. (2018). Automatic detection of traffic lights, street crossings and urban roundabouts combining outlier detection and deep learning classification techniques based on GPS traces while driving. Computers, Environment and Urban Systems, 68, 1-8.

Round 3

Reviewer 1 Report

The paper is well written, there is an experimental part. However, from a technical point of view there is no scientific novelty. The authors simply applied the known technique to new data. Nothing new is offered. As the authors have corrected all the comments, I accept the work, but I ask the editor to appoint another reviewer to maintain the objectivity of the review process and the high status of the journal.